# “Two Cultures in Favor of a Dying Patient”: Experiences of Health Care Professionals Providing Snakebite Care to Indigenous Peoples in the Brazilian Amazon

**DOI:** 10.3390/toxins15030194

**Published:** 2023-03-03

**Authors:** Felipe Murta, Eleanor Strand, Altair Seabra de Farias, Felipe Rocha, Alícia Cacau Santos, Evellyn Antonieta Tomé Rondon, Ana Paula Silva de Oliveira, Hiran Satiro Souza da Gama, Yasmim Vieira Rocha, Gisele dos Santos Rocha, Mena Ferreira, Vinícius Azevedo Machado, Marcus Lacerda, Manuela Pucca, Felipe Cerni, João Ricardo Nickenig Vissoci, Anna Tupetz, Charles J. Gerardo, Ana Maria Moura-da-Silva, Fan Hui Wen, Jacqueline Sachett, Wuelton Monteiro

**Affiliations:** 1Escola Superior de Ciências da Saúde, Universidade do Estado do Amazonas, Manaus 69065-001, Brazil; 2Diretoria de Ensino e Pesquisa, Fundação de Medicina Tropical Dr. Heitor Vieira Dourado, Manaus 69040-000, Brazil; 3Instituto Leônidas and Maria Deane, Fundação Oswaldo Cruz, Manaus 69057-070, Brazil; 4Department of Emergency Medicine, Duke University School of Medicine, Durham, NC 27708, USA; 5Curso de Medicina, Universidade Federal de Roraima, Boa Vista 69310-000, Brazil; 6Instituto Butantan, São Paulo 05503-900, Brazil; 7Diretoria de Ensino e Pesquisa, Fundação Alfredo da Matta, Manaus 69065-130, Brazil

**Keywords:** snakebite envenomations, antivenom, indigenous peoples, access to health care, Brazilian Amazon, intercultural health care

## Abstract

In the Brazilian Amazon, deaths and disabilities from snakebite envenomations (SBEs) are a major and neglected problem for the indigenous population. However, minimal research has been conducted on how indigenous peoples access and utilize the health system for snakebite treatment. A qualitative study was conducted to understand the experiences of health care professionals (HCPs) who provide biomedical care to indigenous peoples with SBEs in the Brazilian Amazon. Focus group discussions (FGDs) were carried out in the context of a three-day training session for HCPs who work for the Indigenous Health Care Subsystem. A total of 56 HCPs participated, 27 in Boa Vista and 29 in Manaus. Thematic analysis resulted in three key findings: Indigenous peoples are amenable to receiving antivenom but not to leaving their villages for hospitals; HCPs require antivenom and additional resources to improve patient care; and HCPs strongly recommend a joint, bicultural approach to SBE treatment. Decentralizing antivenom to local health units addresses the central barriers identified in this study (e.g., resistance to hospitals, transportation). The vast diversity of ethnicities in the Brazilian Amazon will be a challenge, and additional studies should be conducted regarding preparing HCPs to work in intercultural contexts.

## 1. Introduction

Snakebite envenoming (SBE) is classified by the WHO as a neglected tropical disease with significant global morbidity and mortality [1]. Annually, an estimated 5.4 million snakebites—half of which are envenomations—occur worldwide, resulting in up to 138,000 preventable deaths and more than 400,000 amputations and/or other permanent disabilities [1]. Over 90% of the burden of SBEs is concentrated in low-resource regions of low- and middle-income countries (LMIC), in particular among indigenous and rural populations [2]. 

In Brazil, an LMIC in South America, roughly half of all snakebites occur in the Amazon region, and SBE mortality/morbidity is higher than the national average [3]. Previous research suggests that this disproportionate burden is largely due to a lack of accessible antivenom in the region [4,5]. Currently, antivenom (when administered within six hours of the snakebite) is the only effective treatment for SBEs [6,7,8]. Antivenom, however, is limited to urban hospitals, and for most snakebite patients, the journey involved in order to reach treatment can be very difficult [9,10]. 

Indigenous populations in the Brazilian Amazon experience additional inequities related to SBEs [11,12,13]. Compared to the general population, indigenous peoples are 7.5 [9] to 11 [14] times more likely to experience an SBE and 3.5 times more likely to die as a result [15]. Deaths and disabilities from SBEs are a major and neglected problem for this population, which includes approximately 550,000 people distributed in over 100 ethnicities [16]. 

Given the great diversity of indigenous ethnicities, the current literature is not comprehensive, but it does provide some information on how indigenous peoples in the Brazilian Amazon care for themselves and seek traditional care for snakebites [10,17,18]. Minimal research, however, has been conducted on how indigenous peoples access and utilize the health system for snakebite treatment [9]. These different care domains can be conceptualized as the professional (i.e., health system), indigenous or folk medicine (i.e., traditional medicine), and popular (i.e., home remedies) forms of health care [19]. Sectors differ in explaining and treating ill health, defining who is the provider and who is the patient, and specifying how the healer and patient should interact in their therapeutic encounter.

Most studies—like those focusing on traditional medicine for snakebites—tend to examine one sector at a time [19]. In reality, these sectors often overlap. This study sought to understand care pathways for indigenous snakebite victims in terms of the interaction between (1) indigenous patients and professional sector providers (who are almost always non-indigenous) and (2) indigenous and professional sector providers. We invited health care professionals (HCP) to participate in focus group discussions and analyzed their experiences through this lens.

## 2. Results

### 2.1. Characteristics of the Participants 

A total of 56 health care professionals (HCPs) participated in the FGDs (Table 1). Twenty-seven participated in Boa Vista, most of whom were female (*n* = 16, 59%) and nurses or nursing technicians (*n* = 23, 85.2%). Twenty-nine participated in Manaus, roughly half of which were male (*n* = 15, 52%), and all were either nurses (*n* = 18, 62%) or physicians (*n* = 11, 38%). No participants were indigenous. 

Our thematic analysis resulted in six themes and eighteen subthemes (Table 2). The two themes with the most subthemes—health logistics within the professional sector and conflicts between the professional and indigenous sectors—were related to the obstacles involved in providing biomedical care to indigenous peoples. Two themes focused on the perceptions of HCPs, more specifically, the general perceptions of indigenous peoples and the perceptions of indigenous peoples in relation to the acceptability of professional care. One theme was largely descriptive and provided information on typical indigenous care pathways for SBEs. The last theme encompassed recommendations from HCPs for improving indigenous patient care.

### 2.2. HCPs’ General Perceptions of Indigenous Peoples 

Some categories referred to tense or dangerous situations resulting from language problems or cultural clashes, some of which interfered with care delivery.

*“I had to attend to a snakebite patient and I needed to travel by boat to the community. Another indigenous person who was nearby took possession of our boat because he wanted to be a boatman and told me that if I got on the boat without him, he would shoot me.”* (AM, FGD1, P1)

Most HCPs expressed respect and/or admiration for the indigenous peoples they have worked with and/or treated. No HCPs had negative perceptions of indigenous peoples. Any frustrations or grievances that were aired mainly lay within the professional health sector or arose from conflicts between the indigenous and professional sectors. 

*“Yes, they are really strong, I’ve never… I’ve never seen a human being as strong as the indigenous.”* (AM, FGD2, P3)

*“They are wonderful, right? They (the Pirahã) are an ethnic group, a people that you fall in love with. I think that all professionals, if they could spend at least two months there, I think it would be enriching. It would even change your point of view of life.”* (AM, FGD2, P2)

### 2.3. Care Pathways at the Intersection of the Professional and Indigenous Sectors

HCPs described common care pathways for indigenous snakebite patients (Figure 1). Most patients seek traditional medicine before biomedicine, and they often call UBSI or the base hub to come and deliver care in their communities rather than travel to health facilities.

*“If you look at the flowchart in 90% of the communities, the care flowchart would be… first it is traditional treatment at home. Second, [traditional treatment] didn’t work out, so he goes to look for a pajé in the villages that have a shaman; the shaman is the pajé. And then he goes to look for the pastor, and, as a last resort, he goes to the base hub.”* (AM, FGD3, P8)

One HCP provided an illustrative example of an indigenous patient care pathway (Figure 2). A Hupda man was bitten by a snake and moved to a shelter in the forest, far from the village. The HCP explained that patients are often moved deeper into the forest, because many ethnic groups believe “the spirits are good inside the woods.” This patient’s village was remote, a fact that the HCP believes discouraged the indigenous people to seek timely biomedical assistance in addition to the preference for traditional medicine. When the patient’s condition became severe, the health team was contacted. This occurred 12 days post-bite. When the HCPs arrived, the indigenous people helped them transport the patient by hammock to the river, walking for four hours. The team then traveled by boat for one hour to the base hub before transporting the patient to the hospital.

Many HCPs mentioned feeling frustrated at the preference for traditional medicine in situations in which it delays the delivery of biomedical care. Some HCPs referred to traditional medicine as “crazy remedies,” but all HCPs were highly respectful of indigenous decision making and emphasized consent before treating. 

*“Many times, it’s agonizing. You see, you have to wait, but it’s that eternal agony, you’re like ‘I want to help, I want to use my medicine, the medicine that we studied,’ but we have to respect them [their practices].”* (AM, FGD4, P7)

### 2.4. Health Logistics within the Professional Sector

All HCPs described health logistics issues within the professional sector, mainly transportation, human and medical resources, communication, and infrastructure of the UBSIs and base hubs. Transportation and communication were identified as the most significant problems.

*“I’ve had situations where we transported a patient two days after the bite, and sometimes it’s not because of [the patient] refusing [to leave the community], it’s not because of cultural issues, as colleagues mentioned. It is very much because of the logistics.”* (AM, FGD3, P8)

HCPs stressed the majority of indigenous snakebite patients live in remote, rural regions of the Amazon. Most argued that the UBSIs and base hubs do not have sufficient transportation resources, in particular speedboats, aircraft, and fuel, to transfer patients to hospitals with antivenom.

*“It’s difficult for us to win over the indigenous person so that they can come to the city for treatment, but then when you spend two, four, six hours [in the community] to win him over, when you win, you [submit the] request transport. And transport is only aerial. It takes five days of travel to get there in the flood season and seven days in the dry season, so transport is only by air. And then, when I manage to get an indigenous person to go, he has to take the whole family, everyone together… And then [the aircraft] takes two or three days to get here and [waiting] makes them give up.”* (AM, FGD2, P8)

Some HCPs described situations in which they bartered, or “exchanged favors,” with other DSEIs and/or the Brazilian military in an effort to transport patients. Bartering was also discussed in terms of medications, such as analgesics or antibiotics, but given the scarcity of medical supplies, this was less common. HCPs explained that they are more protective of their medications because they do not know when their UBSI or base hub will be resupplied, and a future patient could need that medication.

*“Nearby units either don’t have it (medicine) or say they don’t have it, because [if] you have your patient and you give [the medicine] to my patient, [you could] lose yours. So, they don’t want to give it to us, because the moment it happens to them... And nobody knows the moment, they won’t have [the medicine].”* (AM, FGD4, P4)

Furthermore, due to the lack of antivenom outside hospitals, several HCPs reported using lyophilized antivenom from Colombia (not authorized in Brazil) to treat patients. This lack of medical supplies is partly due to the infrastructure of the UBSIs and base hubs. The HCPs discussed issues regarding the physical structure of their unit, such as a lack of power and the Internet, which delays resupply requests as well as communication with other health units and indigenous communities. 

*“The base hub where I work still has the Internet, for now, but sometimes it goes a week without it, it goes five days without it… To work properly, you send the notification immediately after use [of a medication] and then you receive the replacement, right? And [without Internet]... this can be much slower.”* (AM, FGD1, P5)

Some HCPs worked in UBSIs that did not exist prior to them starting there. These HCPs constructed their units with canvas tents. In cases in which indigenous patients needed to remain under observation, most HCPs described two problems with their unit. There is 1) not enough room or a comfortable place for patients (and their families) to stay, or 2) not enough food to feed both the patients and the HCPs. A few HCPs mentioned sacrificing some of their food supplies to feed patients and/or no places to store the speedboats. 

*“We don’t have a room for the patient to be admitted. We only have the support house, and sometimes we don’t even have a place for the patient to lie down, well-accommodated. I’ve had to cover the floor with a sheet or a simple towel.”* (RR, FGD1, P5)

In addition to transportation challenges, scarce medical resources, and precarious infrastructure, several HCPs raised the issue of human resources. There are not enough HCPs to cover the indigenous population in the Brazilian Amazon. HCPs end up working long hours and often outside their scope of practice. 

*“That’s what I was going to ask (name of other participant), if she has a life, because even when I’m off duty, I can’t have a life. Because, unfortunately, we are all they have.”* (AM, FGD1, P3)

*“We are primary care workers. We are not authorized to do procedures other than primary care, but sometimes we do it because there is no one else to do it.”* (RR, FGD1, P8)

Most HCPs also discussed the risks associated with working in the Brazilian Amazon. The most common risk mentioned was suffering a snakebite, as there is no antivenom available locally. A few participants suggested using the lyophilized antivenom from Colombia. Another common risk was navigating the forest at night. 

*“In addition to the risks of being bitten by snakes, by jaguars, or other accidents that can happen, that we walk under, the ones in the forest, we also have risks at night. Because at the [Base] hub, they always… knock on the window, or when we are sleeping in the shack they (indigenous peoples) knock on the door and call from afar. They keep calling my name, right? Then we leave at any time [and have to travel at night]… We are already sleeping in our uniform for when [the villager] calls. We already know.”* (AM, FGD2, P3)

The last major challenge of health logistics cited by the HCPs was communication. Most HCPs use radio to communicate with each other and with indigenous communities due to the lack of telephone and Internet coverage in the Amazon. Communication includes indigenous communities notifying HCPs of snakebites and providing case details, as well as HCPs exchanging information and discussing the clinical management of patients with other HCPs. Specific limitations noted for radio communication were signal instability during storms (a common occurrence) and narrow operational hours during the day. 

*“Currently, our biggest difficulty is the lack of communication, because in the indigenous region there are some communities that have Internet, but not all of them, and our only means of communication is community radio. There are communities that don’t even have community radio and communication gets very complicated when a snakebite occurs. It takes a long time for the population to be able to ask for help and then get help for a patient”* (RR, FGD1, P7)

### 2.5. Conflicts between the Professional and Indigenous Sectors

Several participants described difficulties arising from the misalignment between official indigenous and health system borders. HCPs on the Roraima–Amazonas border, for example, stated that they are limited to providing care within their state, but face challenges in communicating this to indigenous groups whose villages span the border. 

*“Yanomâmi territory does not see this issue of the geographic area of the states, but that makes a difference for us. So, if there is a case in the Amazonas, and I have the serum here, I cannot say [to the indigenous people] that I will not give the serum because the serum is from Roraima and Amazonas has to give the serum”* (RR, FGD2, P2). 

Furthermore, within states, HCPs discussed situations in which indigenous snakebite patients were reluctant to seek care if the health unit was not on their land. A HCP gave a specific example in which a snakebite patient from an Apurinã village would not go to the base hub, because it was on Paumari land, and the patient died as a result. 

The majority of HCPs emphasized that establishing trust with communities was vital to delivering care. Without trust, HCPs felt it was less likely for communities to contact them and accept biomedical care. 

*“You have to create a bond, because if you don’t create a bond, they won’t go to you. That’s why I went back to my base hub.”* (AM, FGD3, P2)

HCPs raised three specific barriers to creating bonds. The most commonly mentioned issue was short rotations and high staff turnover within DSEIs. Moving from location to location, according to HCPs, ruptures relationships between HCPs and indigenous communities. HCPs noted this was especially problematic among ethnic groups of recent contact, such as the Pirahã, who do not have as deep an understanding of biomedicine. 

*“Yeah, and they don’t let the team in there, all of a sudden, because of the bond, the great staff turnover within the DSEI, right? In the villages, today it is me, tomorrow it can be someone else, the day after tomorrow it is someone else.”* (AM, FGD3, P1)

The second barrier was a lack of antivenom in the UBSIs and base hubs. Without antivenom, HCPs stated that their role is largely basic first aid and mediating the transfer to a hospital, the latter of which most patients do not accept. HCPs felt that they cannot provide effective treatment and, as a result, lose the trust of the indigenous communities.

*“What I wanted to convey is the importance of having the snakebite antivenom, of us having a [positive] solution. We need to be effective, because if he sees that I treated that person and that person died by accident, they won’t believe me anymore, understand?”* (AM, FGD3, P3)

Similarly, the majority of HCPs expressed feeling powerless without antivenom, and that the indigenous communities blame them for poor outcomes in snakebite patients. Some HCPs stated that they also blame themselves, because they consider themselves responsible for indigenous health. 

*“The antivenom was four hours away by speedboat, but it all happened at night and the guy died there. We have nothing to do, right? What are we going to do? We can’t get out of there. It’s very far and then they start to pressure us. The people say ‘But you don’t provide assistance.’ They really blame us, [and say] ‘You let my son die, you let my father die.’”* (AM, FGD2, P4)

Lastly, the third barrier to establishing bonds with communities was the lack of professional preparation for interacting with different indigenous ethnic groups. 

*“23 ethnicities, so 23 different cultures…The language is different, and for me, so far, I couldn’t get all the ethnicities... So, we have 23 difficulties in interacting, precisely because work planning doesn’t account for this and I get something new every year, every month.”* (AM, FGD3, P1)

*“I think we all do this. When we come in, we study the epidemiology of the place, the behavior… even for our clothes. Sometimes, I ask my bosses, ‘Where am I going to go?’ He says, ‘Does it matter?’ I say yes, because, depending on the place, even our clothes, what we are going to take is different.”* (AM, FGD1, P1)

It was emphasized that no DSEI is the same, and within each DSEI, different ethnicities pose different care situations for the HCPs to navigate. 

*“There are ethnicities that are more imposing, aggressive, violent, others are not. There are ethnicities that attribute, for example, illness or death to the team. There are ethnicities that do not even seek care. There are ethnic groups that trust their traditional knowledge more than the team, no matter how much we’ve done there, you know? It’s as diverse as you can imagine.”* (AM, FGD1, P5)

To illustrate their point, HCPs described indigenous perceptions of gender and how that affects their teams’ ability to provide care. HCPs explained that some ethnicities in their region prefer male HCPs, because women who are pregnant, menstruating, or had recent intercourse are considered impure and might worsen the condition of the patient and result in death. 

*“This [patient] was a Hupda. Members of this indigenous ethnicity do not accept a woman touching him if he has been subjected to a snakebite, because, in their culture, if a woman is menstruating, she is unclean and he will die.”* (AM, FGD3, P3)

Other ethnicities, according to the HCPs, do not have these beliefs about impure women, but are intensely patriarchal societies and can pose a threat to female HCPs. 

*“When the nurse is a woman, they want to intimidate. When it’s a team of men, they put themselves in their place; but, when it’s a team of women, they want to... really intimidate.”* (AM, FGD1, P3)

HCPs described other patriarchal societies who do not pose a threat to women but prefer female clinicians, because it is considered unacceptable for male clinicians to touch indigenous female patients. Some HCPs also provided examples of ethnicities who were highly respectful of female clinicians, going as far as to build them houses for privacy while they were treating snakebite patients in their communities. 

Further, largely because of this diversity, the majority of HCPs felt unprepared to work in indigenous health and to treat snakebite patients. Most said they learned by doing things day to day from the HCPs who arrived before them and the indigenous communities themselves. 

*“My first experience was... I was more scared than the patient, because I had arrived at the DSEI in the base hub and the patient had been bitten. He didn’t present anything, super mild symptoms, just a local pain, and I was desperate without knowing what to do.”* (AM, FGD4, P5)

*“I ended up learning to deal with everything, but the first time I was desperate, because you have a patient in your hands and you look at the nursing technician and the technician is you. [Then], the second time, more or less, the third time, it’s already becoming routine.”* (AM, FGD4, P4)

### 2.6. HCPs’ Perceptions of Indigenous Amenability to Professional Care

All HCPs agreed that most indigenous communities are receptive to antivenom administration but highly reluctant to leave their villages. 

*“They always arrive with some herbs on the wound. There are a lot of snakebite cases at the clinic because the indigenous people already know that it is a serious thing and that their traditional medicine alone will not solve it. Need to accept western medicine.”* (RR, FGD2, P5)

*“They even found out I was here, and the indigenous people called me, right? Asking if I was going to come in with the antivenom. It’s because they don’t want to come to the city. They don’t like the cultural clashes of coming to the city.”* (AM, FGD2, P4)

The HCPs discussed several reasons for which the indigenous patients refuse to be transferred to hospitals, including, but not limited to, spiritual beliefs regarding reincarnation, fear of death, fear of amputation, and cultural food preferences (Table 3). 

### 2.7. HCPs’ Recommendations for Improving Professional Care of Indigenous Patients

Another significant reason for refusing hospital treatment is the inability to concurrently receive traditional treatment. The majority of HCPs emphasized—and recommended—a joint approach to treating snakebite patients within their communities for more effective patient care and positive outcomes. 

*“I get along very well with the shamans, because I’ve been working with the ethnic group (Huni Kuin) for two years. So, it’s me and the shamans. We create bonds like this. First, he comes in and does the work. I let him do his work, for as long as he needs. Then he goes and says, ‘Doctor, now it’s up to you.’ … And today, what interferes the most is that they don’t want to leave the... the village, but other than that, my work and the shamans’ work, my work after the shaman leaves doesn’t interfere with anything.”* (AM, FGD2, P8)

*“We have to respect it because, for patients, it’s a matter of their millennia of tradition and knowledge. A Yanomâmi girl who we treated for snakebite also wanted to call the shamans and healers. And, in this situation, we had to talk and negotiate to perform our treatment along with their treatment. We had to prove that what we were doing with the patient would benefit him and that what the shaman and the healer were doing would also benefit him. In other words, we united the two beliefs—two cultures in favor of the health of a dying patient.”* (RR, FGD1, P5)

However, in order to deliver effective biomedical care alongside traditional treatments, the HCPs stressed the need for antivenom. Decentralizing antivenom to UBSIs and base hubs would likely avoid complications and reduce the need to transfer patients to hospitals.

*“We are fighting for this to become a reality, for it to really happen, so that we can help, so that we can have a faster action, more resolution, in the area where I work… The valley is immense. We have several villages, [and] we’ve lost many patients because there was no plane. Imagine if you go to a village up there and take a boat for ten days, for six days. It’s complicated, so we really need this support in the area to be able to have a better result regarding the cases of ophidian accidents.”* (AM, FGD4, P7)

*“We have lost children, so this is exactly our greatest anguish. So, when this possibility of decentralization occurs, it is wonderful for us, because we know that if I have a vial of antivenom in (name of region) and [a snakebite] happens… I can do the logistics. But if there is no antivenom in (names of regions), and only in Manaus, then it is difficult.”* (AM, FGD4, P4)

## 3. Discussion

This study analyzed the experiences of health care professionals (HCP) treating indigenous patients and interacting with indigenous caregivers to understand care delivery for indigenous SBE patients. Our thematic analysis highlighted three key findings: (1) indigenous peoples are amenable to receiving antivenom but not to leaving their communities to go to hospitals; (2) HCPs require antivenom and additional resources in UBSIs and base hubs to adequately treat patients; and (3) HCPs strongly recommend a joint, bicultural approach to treating indigenous SBE patients.

To our knowledge, there is no literature regarding indigenous amenability to antivenom in the Brazilian Amazon [9]. HCPs’ perspectives were that the indigenous communities were not against biomedical SBE services, especially antivenom and, in fact, expected this care in order to build up the trust in the HCPs and the health system. Two independent studies on other indigenous health issues in the Brazilian Amazon indicated indigenous acceptability of biomedicine and support this finding. In the Vale do Rio Javari, about 80% of indigenous people reported utilizing both indigenous and biomedicine to treat chronic pain [20]. In the Alto Solimões, almost all indigenous people were amenable to point-of-care screening for syphilis and HIV, and 87% of those who tested positive received biomedical treatment [21]. 

A central obstacle to indigenous peoples’ receiving biomedical treatment, according to the HCPs, was resistance to leaving their communities and traveling to hospitals. This is an important result, because it emphasizes the need to have professional treatment available in the indigenous villages in order to avoid delays and unnecessary deaths from resistance to hospitals. The Alto Solimões study also reported this finding [21], in addition to HCPs working in an Indigenous Health Center (CESAI; a higher-level outpost than base hubs and often located in cities) in Santarém, Pará, a small city on the Amazon River. These HCPs also cited fear or anxieties about receiving care in a city, poor unit infrastructure, and cultural food preferences as reasons for the indigenous patients’ reluctance to leave [22]. 

The second major finding was that of HCPs requesting additional infrastructure and medical and human resources to improve patient care. Again, the Alto Solimões study complemented our findings, describing a lack of fuel, speedboats, and aircraft; lack of medical supplies; and high staff turnover and increased workload in their DSEI [21]. A different study regarding the highly nomadic Yanomami people discussed the lack of transportation resources as a major obstacle to accessing professional care [23]. Mendes et al. (2018) performed a functional analysis of the Indigenous Health Care Subsystem and confirmed these findings, stating that “discontinuity of care, added to the shortage and high turnover of professionals,” challenges the effectiveness of the system [24]. 

To work around some of these logistical challenges, our HCPs specifically advocated for additional training and access to antivenom. Many pointed out that their UBSIs or base hubs are capable of storing antivenom in the refrigerators for vaccines. Lack of electricity, though a problem in the past, is less applicable now with the increasing use of solar energy for refrigeration in the Brazilian Amazon [25]. A second and previously relevant obstacle in antivenom decentralization was the potential for adverse reactions to antivenom, which are milder and less frequent with improvements in antivenom manufacturing [26]. A recent resource-mapping analysis of SBE care in the Brazilian Amazon recommended decentralizing antivenoms to primary health units for both indigenous and non-indigenous populations in order to avoid the financial and emotional cost of transferring SBE patients to urban hospitals and to improve patient outcomes [27]. 

Regarding additional training for HCPs, a clinical practice guideline has been recently developed and validated for training primary health workers in the Brazilian Amazon on the clinical management of snakebites and in antivenom administration, including how to treat possible adverse reactions [28]. The HCPs, as well as several studies, advocated future training initiatives or system strengthening efforts that should also include caregivers from the indigenous sector [9,24,29,30]. Herndon et al. (2009) expressed the need for integration, explaining that indigenous peoples in the Brazilian Amazon are “caught between an undervalued and disintegrating traditional culture and an inaccessible western system” [29]. The integration process itself should be co-led with indigenous providers, and such an effort could be a potential intervention to improve indigenous SBE patient outcomes [31,32,33,34]. 

Brazil is internationally recognized for its laws and policies aimed at improving the health of indigenous peoples [35]. The National Health Policy for Indigenous Peoples built a complementary and differentiated model of health services specifically to protect, promote, and restore indigenous health. This indigenous health system must, following Brazilian legislation, respect indigenous cultures and promote the articulation of traditional indigenous health systems with biomedicine [24,30]. The HCPs who participated in the study asserted that a joint, cross-cultural approach is likely the most effective and efficient treatment of snakebite envenomations, but lack of training and resources—most notably antivenom—restricts success. The need for integrated professional and indigenous care is not limited to snakebites. For example, three studies on tuberculosis treatment for indigenous peoples in the Brazilian, Peruvian, and Colombian Amazon explicitly recommend an integrated care approach [36,37,38].

Our findings, though consistent with the literature on biomedical care and indigenous populations in other disease areas, are particularly striking given that snakebite envenomations are an emergency health condition with freely available efficacious treatment. The Brazilian health system provides antivenom at no cost to the population [27]. SBEs are highly treatable, especially as HCPs reported most indigenous peoples accept and even expect antivenom. This disease burden on the indigenous population is entirely preventable. The central obstacles to overcome in improving SBE patient outcomes were the patient’s need to seek higher-level care and, thus, leave the indigenous village, alongside the lack of resources that strangle indigenous health care as a whole. Though systemic challenges are complex and multifaceted, decentralizing antivenom from hospitals to the local base hubs and UBSIs is feasible, likely cost-effective, and addresses these central obstacles [9,27]. 

This study is limited in that it does not represent the experiences and perceptions of all the HCPs working with indigenous peoples in the Brazilian Amazon. Despite efforts to minimize interference, the focus group discussions might contain generalizations from participants, in particular when reporting the behavior of other people. The perceptions of HCPs that were collected might not correctly reflect the reality and cultural practices of the indigenous ethnicities mentioned. These HCPs were not indigenous, and many were not originally from the Amazon region. Their point of view is rooted in western education and health systems and could be biased in its favor. Lastly, ~15% of the audio recordings from the Boa Vista, Roraima focus group discussions were inaudible due to a technical issue with the recorders.

## 4. Conclusions

This study analyzed the experiences of health care professionals (HCPs) that treat indigenous patients and interact with indigenous caregivers to understand care delivery for snakebite envenomations (SBEs) that involve indigenous patients. Our thematic analysis highlighted that most indigenous peoples are amenable to receiving antivenom but not to leaving their communities to go to hospitals. Consequently, the HCPs asserted that a joint, cross-cultural approach is likely the most effective and efficient SBE treatment for indigenous patients, but lack of training and resourcdes—most notably antivenom—restricts success. Future studies should engage with indigenous populations, both patients and caregivers, in order to ascertain their experiences and develop recommendations for improving SBE treatment. Decentralizing antivenom from hospitals to the local base hubs and UBSIs addresses the central obstacles identified in this study (e.g., resistance to hospitals, transportation) and is feasible and likely cost-effective. The vast diversity of ethnicities in the Brazilian Amazon will be a challenge, and additional studies should be conducted regarding how to best prepare HCPs to work in intercultural contexts as well as approaches for integrating the indigenous and professional sectors in an antivenom decentralization strategy.

## 5. Materials and Methods

### 5.1. Study Design

An exploratory descriptive study was conducted in order to understand the experiences of health care professionals providing biomedical (i.e., modern, western) care to indigenous peoples with snakebite envenomations in the Brazilian Amazon. Focus group discussions (FGDs) were performed in the context of a three-day training workshop for health care professionals (HCPs) who work for the Indigenous Health Care Subsystem [39]. The training focused on a recently developed clinical practice guideline for treating SBEs [28]. A qualitative thematic analysis was conducted with a primarily inductive approach [40]. Emerging themes and subthemes were interpreted using Kleinman’s model of the three health sectors [19].

Participant recruitment started in September 2021. FGDs occurred in Boa Vista, Roraima (13–15 October 2021) and Manaus, Amazonas (27–31 July 2022). Data analysis was concluded in January 2023. This study was reported according to the Consolidated Criteria for Reporting Qualitative Research (COREQ) [41] (see Appendix A). 

### 5.2. Study Setting

Boa Vista (Roraima) and Manaus (Amazonas) are cities in northern Brazil that are geographically part of the Amazon Rainforest. Training workshops and FGDs were conducted in these cities, since the states of Amazonas and Roraima have the highest snakebite incidence in Brazil, an estimated ~62 and ~65 per 100,000 inhabitants, respectively, and the largest indigenous populations [14,42]. Amazonas has the most indigenous ethnicities and thus the highest number of HCPs providing care, and almost half of Roraima is indigenous land [43]. 

Care for indigenous peoples is provided by the Indigenous Health Care Subsystem, which operates within the national universal health care system [44]. This subsystem organizes care delivery via special indigenous health districts (Distrito Sanitário Especial Indígena; DSEIs) [43]. There are nine DSEIs within our study setting, two in Roraima and seven in Amazonas (Figure 3).

The function of DSEIs is to provide primary care on indigenous land, and they serve as an entrance point to the health system [44]. Within the DSEIs, there are two types of health facilities: the basic units of indigenous health (UBSI) and the base or health hubs. UBSIs are primary health facilities near or within indigenous villages and often the first point of contact for patients. If a patient requires specialized attention, HCPs refer them to the closest easily accessible base hub. HCPs at the base hubs then provide care and/or organize transfers to urban hospitals [9]. 

In terms of snakebite envenomations, UBSIs and base hubs have a similar role. Antivenom is only available in hospitals, so the HCPs in the DSEIs provide first aid, such as wound cleaning and analgesics, and facilitate transport of indigenous patients to the hospital [9]. The health care landscape of the nine DSEIs in the study is further detailed in Table 4. 

### 5.3. Research Team and Reflexivity

The research team consisted of one female PhD-level researcher and one male PhD-level researcher with extensive experience in clinical research and care of SBE patients (J.S., W.M.), two male PhD-level qualitative researchers with experience in neglected tropical diseases (F.M., V.A.M.), a male MSc-level researcher and professor of Indigenous Health (A.S.d.F.), a female MSc-level qualitative researcher and doctor of physical therapy (A.T.), one male and one female BS-level qualitative researcher (F.R., E.S.); several graduate students at the Fundação de Medicina Tropical Dr. Heitor Vieira Dourado, a tertiary hospital that treats SBE patients in the western Brazilian Amazon, participated as observers during the FGDs. The FGD guide was developed by J.S., W.M., F.M., V.A.M., and A.S.d.F. FDGs were led by W.M., F.M., V.A.M., A.S.d.F., F.R., and M.F. Data analysis was conducted by W.M., F.M., F.R., M.F., and E.S. The study team had no prior relationship with the participants. One member of the team is indigenous (A.S.d.F.). W.M., F.M., and A.S.d.F. have previously researched professional and traditional SBE care in indigenous populations in the Brazilian Amazon. 

### 5.4. Recruitment

WM and MP contacted the DSEI coordinators (*n* = 9) within the states of Amazonas and Roraima, respectively, by phone and email. The DSEI coordinators then summoned their nursing technicians, nurses, and physicians who were available for a three-day training workshop (one in Roraima, one in Amazonas) on a clinical practice guideline for treating snakebite envenomations [28]. Attendees were invited by the study coordinator (WM) to participate in the FGDs.

### 5.5. Focus Group Discussion Procedure

The interviewers used a semi-structured FGD guide with three open-ended questions and several follow-up questions, which allowed the interviewers to probe participant responses and investigate the participants’ perceptions of their experiences with snakebites suffered by indigenous patients (Table 5). FDGs were performed in Portuguese in a comfortable and quiet room with the mediator, observers, and participants. Discussions lasted an average of one hour. Field notes were recorded by the mediator and by observers for data triangulation. A total of eight FDGs were carried out with approximately eight health professionals each, one experienced mediator, and two observers. Inductive thematic saturation was achieved in the analysis when no new themes or patterns emerged from the collected data, indicating that sufficient data had been gathered to understand the research topic.

### 5.6. Data Analysis

FGDs were recorded, transcribed (ER, AO, HG), de-identified, and inserted into the MAXQDA 20 program. The qualitative thematic analysis was carried out with the elaboration of themes identified during inductive coding of the transcripts [40]. These themes and emerging subthemes were discussed among the researchers (F.M., A.S.d.F., F.R., M.F., E.S., and W.M.). Differences that emerged in the coding process were resolved through discussion. Transcript data were triangulated with observations and field notes to validate the results and ensure a reflective analysis of the data [45]. Following the inductive thematic analysis, FM and WM interpreted the themes and subthemes via Kleinman’s model of the three health sectors [19], focusing on the strengths, weaknesses, and interconnections of the indigenous and professional sectors and identifying opportunities for improving snakebite envenoming care. 

Although the analytics research team was experienced in qualitative research on SBEs, they had not yet interacted with these health professionals. Data were reviewed and discussed with an indigenous health expert (A.S.d.F.) to mitigate interpretation biases.

Representative quotes chosen for this manuscript were translated from Portuguese to English by bilingual research team members (F.R. and F.M.).

## Figures and Tables

**Figure 1 toxins-15-00194-f001:**
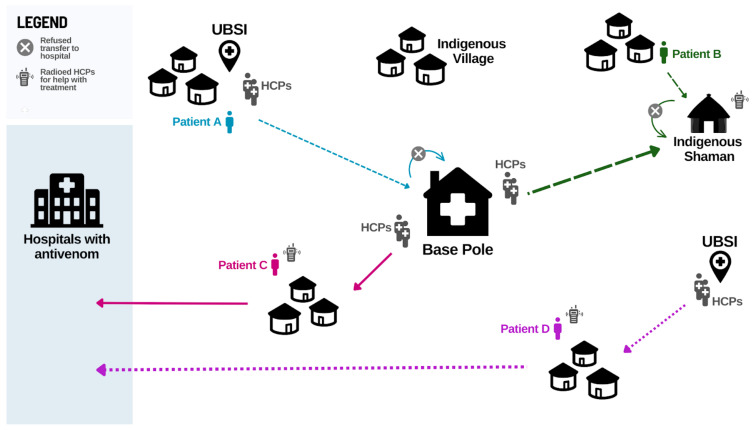
Main snakebite care pathways described by HCPs. An indigenous village calls the nearby UBSI team for help immediately after the SBE. The team and patient’s family carry the patient to the base hub, where they remain for three days with their family. An indigenous villager seeks care from the shaman after a snakebite. When the patient does not improve, the village calls the base hub. The base hub’s team takes a speedboat and then hikes through the forest to reach the village and stays there for two days to provide care alongside the shaman. The base hub’s team is called to an indigenous village four days after a snakebite. Traditional medicine and home remedies did not improve the patient’s condition. The patient has a severe case, and the team, patient, and patient’s family travel by speedboat to a hospital. An indigenous village calls the UBSI team immediately to help a pediatric snakebite patient. The team travels by speedboat to the village and provides basic care. After two days, the patient is not improving, and the family accepts transport. The village is remote, and it takes an additional day to organize aerial transport to the hospital.

**Figure 2 toxins-15-00194-f002:**
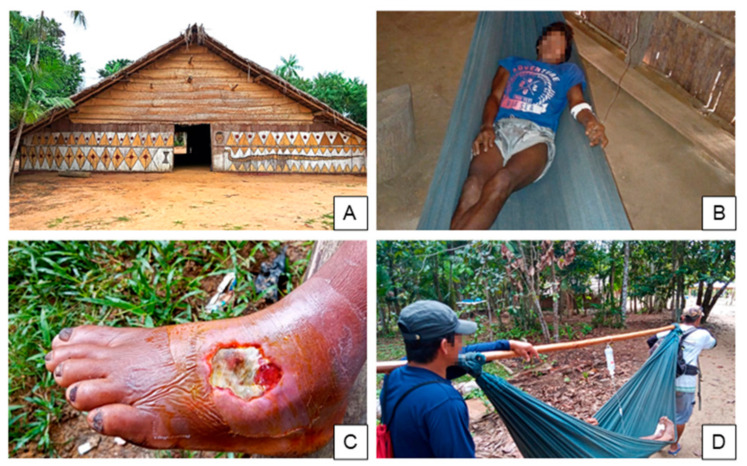
A case example of a care pathway for an indigenous snakebite patient. (**A**) A typical indigenous structure for patient accommodation duri›ng disease treatment and recovery in the forest. (**B**) Male SBE patient resting in the structure and undergoing traditional treatment. (**C**) Ulcer from snakebite observed in the left foot of the same patient, 12 days post-bite. (**D**) HCPs transporting the patient using a hammock to the base hub and then to the hospital to receive treatment.

**Figure 3 toxins-15-00194-f003:**
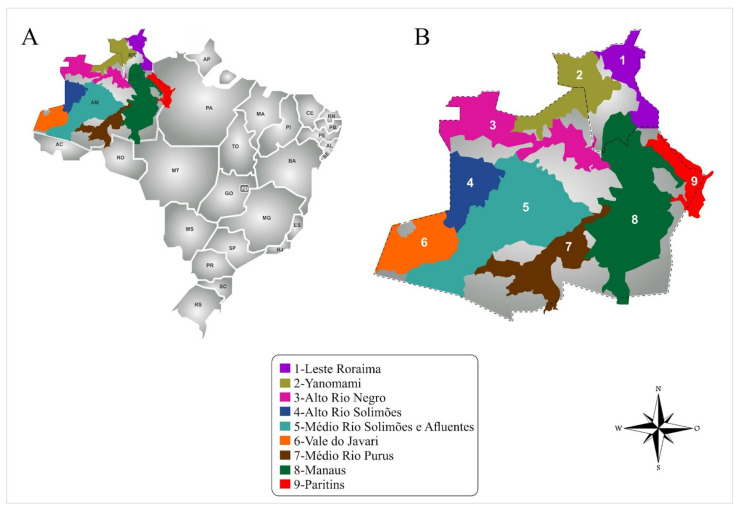
Study setting. (**A**) Brazilian territory; (**B**) Nine special indigenous health districts (DSEIs), two in Roraima and seven in Amazonas, where participants were recruited.

**Table 1 toxins-15-00194-t001:** Characteristics of the participants in the FGDs held in Boa Vista and Manaus.

Variable	Participants from Boa Vista, Roraima (*n* = 27) ^1^	Participants from Manaus, Amazonas (*n* = 29)	Total Participants (*n* = 56)
Sex (male)	11 (41%)	15 (52%)	26 (46%)
Profession			
Physician	2	11	13
Nurse	14	18	32
Nursing technician	9	0	9
Veterinarian ^2^	1	0	1
Biologist ^2^	1	0	1
Ethnicities named in transcripts	Apurinã, Baniwa, Baré, Kokama, Hixkaryana, Huni Kuin, Hupda, Hupdes, Kanamari, Korubo, Kulina Pano, Macuxi, Marubo, Matis, Matsés, Paumari, Pirahã, Sateré-Mawé, Suruhuarrá, Tariano, Tenharim, Tsohom-dyapa, Tukano, Yamamadi, Yanomami		

^1^ Two participants from were excluded this analysis. One participant did not have experience with snakebite envenomations, and one participant’s contributions were inaudible in the recordings. ^2^ These participants have formal training as a veterinarian and as a biologist, but also have professional experience in the health care field and with transportation logistics of patients from remote areas.

**Table 2 toxins-15-00194-t002:** Summary of emergent themes and subthemes from the FGDs.

Themes	Subthemes
1. HCPs’ general perceptions of indigenous peoples	
2. Care pathways at the intersection of professional and indigenous sectors	2.a. HCPs’ perceptions of indigenous medicine for SBEs 2.b. Narrative of the care pathways of indigenous patients
3. Health logistics within the professional sector	3.a. Lack of transportation resources 3.b. Communication network failures 3.c. Insufficient medical supplies 3.d. Lack of human resources 3.e. Poor infrastructure of health facilities 3.f. Health risks or personal risks to HCPs
4. Conflicts between the professional and indigenous sectors	4.a. Issues arising from the misalignment between official borders 4.b. Importance of trust with indigenous communities 4.c. Short rotations and high turnover 4.d. Inability of HCPs to provide effective treatment 4.e. Lack of intercultural training for HCPs 4.f. Influence of diversity on HCP care delivery
5. HCPs’ perceptions of indigenous peoples in relation to the acceptability of professional care	5.a. Indigenous peoples’ amenability to receiving antivenom 5.b. Indigenous resistance to leaving their communities
6. HCPs’ recommendations for improving professional care of indigenous patients	6.a. Joint, bicultural approach to SBE treatment 6.b. Decentralizing antivenom to non-urban health facilities

**Table 3 toxins-15-00194-t003:** Reasons cited by HCPs for indigenous reluctance to go to hospitals that have antivenom available.

Resistance to Hospitals	Illustrative Quote(s)
Spiritual beliefs regarding reincarnation	*“She was coming from the field, but she was on the way, almost arriving home… She died. She came crawling to try to get there alive, desperate, without letting anyone touch her to get home. Why? Because they believe that if they die outside their home, their environment, their land, where they live… they, if they reincarnate, they will come as beings of the forest. They will not come in the afterlife to the physical human body. So, they cannot die outside [their lands] because they want to come back the same way.”* (AM, FGDV, P3)
Fear of death	*“They don’t like to go to the city at all. For them, it is, they think that if they go there, usually when they go it’s because they are already very critical patients, and they often don’t come back. They come back in the coffin. That’s why they’re so afraid to go to the city.”* (AM, FGDM, P2)
Fear of amputation	*“The reason they don’t like coming to the city is that they assimilate the city to the compartment syndrome. So, they say, ‘I don’t want to go. I want to treat myself here, because they’ll take my foot. They’ll take my leg. I’ll come back all cut.’ Our difficulty to get them out of there for treatment is that, because they can’t understand, right, this process… this negative evolution of the snakebite. So, they’re afraid of compartment syndrome, and sometimes they get compartment syndrome because they don’t go, and we don’t get access. The shaman goes there, the shaman sometimes sucks the area, makes a tourniquet, puts everything you can imagine on the skin that makes our work difficult, they do it. And you only have access after all possibilities are over.”* (AM, FGDM, P8) *“[It is] complex for them (amputees) to come back, because they will be rejected. It is as if they have no means of subsistence for themselves, becoming an invalid, and when they get the prosthesis, they also don’t feel well and wonder why they need to use it. And they already imagine that they will suffer rejection in the community, so that’s why they end up not returning to the community anymore.”* (RR, FGDW, P2)
Cultural food preferences	*“Because of these difficulties… we nurses, we got together and went to talk to the municipal health secretary to see if we could make a more cozy environment in the hospital. And we did, [we put] some paintings of the forest [and] put a hammock tie. We talked to the nutritionist at the hospital where the secretary gave support. For example, Pirahã food is just fish thrown in the water and that’s it. If you put black pepper and paprika, they won’t eat it.”* (AM, FGDM, P8)

**Table 4 toxins-15-00194-t004:** Special indigenous health districts in the study setting.

State	DSEI #	Area (km^2^)	Population	Health Care Units	Ethnicities and Villages
Amazonas	Vale do Rio Javari	91,384.29	6082	21 UBSIs *, 7 base hubs	8 ethnicities, 66 villages
	Manaus	303,092.01	31,468	5 UBSIs, 17 Base hubs	48 ethnicities, 263 villages
	Médio Rio Solimões and Tributaries	297,616.37	20,867	19 UBSIs, 15 base hubs	21 ethnicities, 186 villages
	Alto Rio Negro	138,020.94	27,769	5 UBSIs, 25 base hubs	23 ethnicities, 685 villages
	Alto Rio Solimões	79,763.43	70,659	15 UBSIs, 12 base hubs	7 ethnicities, 241 villages
	Médio Rio Purus	105,806.98	8770	7 UBSIs, 10 base hubs	11 ethnicities, 123 villages
	Parintins	50,644.96	16,582	12 UBSIs, 12 base hubs	11 ethnicities, 127 villages
Amazonas and Roraima	Yanomami	106,327.56	29,934	27 UBSIs, 37 base hubs	2 ethnicities, 370 villages
Roraima	Eastern Roraima	69,755.08	55,089	245 UBSI, 34 base hubs	7 ethnicities, 342 villages

Information provided by the Brazilian Ministry of Health, 2022 [43]. # DSEI: Distrito sanitário de saúde especial indígena (special indigenous health district) * UBSI: Unidade básica de saúde indígena (basic units of indigenous health).

**Table 5 toxins-15-00194-t005:** Focus group discussion guide.

Question/Theme	Objective
Experience with SBEs	
Which of you here has seen or treated a person who was a victim of a snakebite? Could you share your experience? Explore answers: → Place where the health care was provided (in the hospital or not); → Severity of cases; → Complications/outcomes; → Use of antivenom; → Traditional treatments and self-medication. Where did the knowledge you used in these treatments come from?	Assess the confidence level of professionals in the diagnosis and treatment of SBEs.
SBE patient itinerary	
How do patients who have been bitten by snakes usually arrive at the health facility where you work? Explore answers: → Means of transport; → Itineraries; → Difficult access—what are the challenges in accessing antivenom; → Severity on admission. → What happens to the patient when he/she leaves the health unit where you work? → Explore answers: → Discharge from the health unit; → Referral to other units; → Means of transport in referring and challenges; → They are left with sequelae and need follow-up—is there follow-up?	Understanding SBEs and follow-up by hospital/facility
Treatment	
What would be the ideal treatment for a snakebite patient? Does it impact your workload? Is there anything that is missing at your current facility in order to provide the best care for patients? If yes, what?	

## Data Availability

Data of this study are unavailable publicly due to privacy and ethical restrictions. Data may be shared with interested parties when requested of the corresponding author.

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
