# Peer review of "“Two Cultures in Favor of a Dying Patient”: Experiences of Health Care Professionals Providing Snakebite Care to Indigenous Peoples in the Brazilian Amazon"

_toxins, 2023, doi:10.3390/toxins15030194_

Round 1

Reviewer 1 Report

This 

This manuscript summarises the challenges of accessing anti-venom and moving indigenous Brazilians to advanced health care. It clearly outlines the issues regarding access and challenges and supports, probably what is the case with most ethnic groups in remote areas globally. I have no doubt it will be a widely read article as it is very well written. I recommend it for publications with a few minor formatting correction and one suggestion as detailed below.

My one suggestion for the manuscript is to acknowledge what may be problems in having anti-venom on site and issues that need to be addressed in conjunction with training HCP’s to administer anti-venom e.g. storage of anti-venom, if refrigeration is required do power blackouts and lack of electricity affect this, anti-venom shelf life etc. Just one or two sentences would reinforce some additional issues that remote communities face. To document some additional challenges will help support guide the development of better most cost effective venoms for remote communities. I think this will lift the paper and make it more citable to a range of researchers.

It is a really significant and nice paper, congratulations to the authors.

Abstract

Second line insert word ”the” prior to word … indigenous population.

Results

Table 4. I find the formatting hard to read. Do the subthemes relate to a specific theme? I would recommend to change the bullet points to numbers align the writing to the top of the cell. i.e. If the subthemes to relate to the broader theme then number sub themes according, it will aid the reader to quickly discern the relatedness. If they do not relate back to a main theme the number the subthemes instead of dot points.

Theme

Subtheme

1. HCPs general perceptions…..

1a. CPS’s perceptions of ind…

1b. Narrative care pathways…

2. Care pathways…

2a. Lack of transport

In all captions and footnotes of Figure captions as there appears to be two full stops at the end of sentences e.g. see Fig 2.

Author Response

This manuscript summarises the challenges of accessing antivenom and moving indigenous Brazilians to advanced health care. It clearly outlines the issues regarding access and challenges and supports, probably what is the case with most ethnic groups in remote areas globally. I have no doubt it will be a widely read article as it is very well written. I recommend it for publications with a few minor formatting correction and one suggestion as detailed below.

My one suggestion for the manuscript is to acknowledge what may be problems in having antivenom on site and issues that need to be addressed in conjunction with training HCP’s to administer anti-venom e.g. storage of anti-venom, if refrigeration is required do power blackouts and lack of electricity affect this, anti-venom shelf life etc. Just one or two sentences would reinforce some additional issues that remote communities face. To document some additional challenges will help support guide the development of better most cost effective venoms for remote communities. I think this will lift the paper and make it more citable to a range of researchers.

It is a really significant and nice paper, congratulations to the authors.

RESPONSE: Many thanks for the comments. Sentences were added to reinforce some additional issues that remote communities face (See paragraphs 5 and 6 of the Discussion).

Abstract

Second line insert word ”the” prior to word … indigenous population.

RESPONSE: The word was added accordingly.

Results

Table 4. I find the formatting hard to read. Do the subthemes relate to a specific theme? I would recommend to change the bullet points to numbers align the writing to the top of the cell. i.e. If the subthemes to relate to the broader theme then number sub themes according, it will aid the reader to quickly discern the relatedness. If they do not relate back to a main theme the number the subthemes instead of dot points.

RESPONSE: Change was made accordingly.

In all captions and footnotes of Figure captions as there appears to be two full stops at the end of sentences e.g. see Fig 2.

RESPONSE: Revision was made accordingly.

Reviewer 2 Report

The present article is a descriptive study about indigenous health care focused on snakebite envenoming. Unfortunately, the low number of individuals interviewed argue reduces the high scientific value of the study. However, it is understandable  the difficult of elaborating and developing the study in the present form. Taking it together, the article is very welcome and of major importance to be publicized, specially at the political moment Brazilian government is working hard to nurse indigenous population, particularly the Yanomamis. I highly encourage the authors to provide a deep English review, if possible with the help of a native speaker, if possible. 

Author Response

The present article is a descriptive study about indigenous health care focused on snakebite envenoming. Unfortunately, the low number of individuals interviewed argue reduces the high scientific value of the study. However, it is understandable the difficult of elaborating and developing the study in the present form. Taking it together, the article is very welcome and of major importance to be publicized, specially at the political moment Brazilian government is working hard to nurse indigenous population, particularly the Yanomamis. I highly encourage the authors to provide a deep English review, if possible with the help of a native speaker, if possible. 

RESPONSE: Many thanks for the comment. A deep English review was made accordingly.

Reviewer 3 Report

The manuscript describes health professionals' experiences (HP) and the provision of medical care for poisoning indigenous patients in the Amazon. The work is well presented and adequately describes the interests and procedures adopted in the face of this problem experienced by an isolated population with certain access restrictions. The subject is complex and involves several sectors of public administration in Brazil. Although the subject's interest is restricted to a small group, it is important to publish the study.

Author Response

The manuscript describes health professionals' experiences (HP) and the provision of medical care for poisoning indigenous patients in the Amazon. The work is well presented and adequately describes the interests and procedures adopted in the face of this problem experienced by an isolated population with certain access restrictions. The subject is complex and involves several sectors of public administration in Brazil. Although the subject's interest is restricted to a small group, it is important to publish the study.

RESPONSE: Many thanks for the comment. We believe that the topic is of interest to toxinologists, epidemiologists, public health scientists, clinical scientists, and health managers.

Round 2

Reviewer 2 Report

I have no further comments